# Correlation between Anti-SARS-CoV-2 Total Antibodies and Spike Trimeric IgG after BNT162b2 Booster Immunization

**DOI:** 10.3390/vaccines10060890

**Published:** 2022-06-02

**Authors:** Gian Luca Salvagno, Giuseppe Lippi

**Affiliations:** Section of Clinical Biochemistry, University of Verona, 37126 Verona, Italy; gianluca.salvagno@univr.it

**Keywords:** COVID-19, SARS-CoV-2, vaccination, antibodies, booster

## Abstract

Objective: In this work we monitored both total and IgG anti-SARS-CoV-2 antibodies responses after BNT162b2 vaccine booster immunization in a cohort of ostensibly healthy healthcare workers. Methods: The study population consisted of 266 subjects (median age, 46 years and interquartile range (IQR), 35–52 years; 168 females) undergoing homologous 30-µg BNT162b2 booster administration. Serum samples were collected immediately before the booster dose and 1 month after. Results: The concentration of anti-SARS-CoV-2 RBD total antibodies and anti-SARS-CoV-2 spike trimeric IgG increased by 31 (IQR, 16–53) and 22 (IQR, 11–43) folds, respectively, after receiving the BNT162b2 vaccine booster. A highly significant Spearman′s correlation was found between the relative increase (i.e., ratio of post-booster and pre-booster serum values) of anti-SARS-CoV-2 RBD total antibodies and anti-SARS-CoV-2 spike trimeric IgG (r = 0.86; p < 0.001). Conclusion: These results suggest that monitoring of post-BNT162b2 booster immunization for purposes of identifying “low responders” could be conducted almost interchangeably with either total or IgG anti-SARS-CoV-2 antibodies.

## 1. Introduction

In a recent article published in this journal, Skrzat-Klapaczyńska and colleagues [1] showed that a booster dose of the mRNA-based BNT162b2 vaccine was effective at increasing the serum values of SARS-CoV-2 IgG antibodies measured with the MAGLUMI SARS-CoV-2 S-RBD IgG assay over the maximum measurable level according to the method used by the authors (i.e., >433 BAU/mL). To provide further insights on this important aspect in terms of limiting SARS-CoV-2 infection and lowering the risk of developing serious complications from coronavirus disease 2019 (COVID-19), we monitored both total and IgG anti-SARS-CoV-2 antibodies response after BNT162b2 vaccine booster immunization in a cohort of ostensibly healthy healthcare workers, with a specific focus on emphasizing the potential correlation between the post-booster variation of these two antibodies classes.

## 2. Materials and Methods

This serosurveillance observational study is based on a cohort of ostensibly healthy workers of Pederzoli Hospital (Peschiera del Garda, Verona, Italy), who volunteered to undergo COVID-19 vaccination with the mRNA-based Pfizer/BioNTech BNT162b2 vaccine (Pfizer Inc., New York, NY, USA; double 30-µg dose with 3 weeks interval), followed by homologous 30-µg booster dose >8 months thereafter. Nucleic acid amplification tests were carried out at 2–4 weeks interval throughout the study period with Altona Diagnostics RealStar SARS-CoV-2 RT-PCR Kit (Altona Diagnostics GmbH, Hamburg, Germany) or Seegene Allplex SARS-CoV-2 Assay (Seegene Inc., Seoul, South Korea) to rule out incident SARS-CoV-2 infections, that would lead to eliminating these subjects from the study, since booster vaccination is not normally given to people with a history of recent infection. Blood was drawn throughout the study period, including the same day as receiving the BNT162b2 vaccine booster and 1 month afterwards. Serum was used for assaying anti-SARS-CoV-2 RBD total antibodies with Roche Elecsys Anti-SARS-CoV-2 S on Roche Cobas 6000 (Roche Diagnostics, Basel, Switzerland), a method which measures total anti-SARS-CoV-2 anti-RBD (receptor-binding domain) antibodies [2], as well as with anti-SARS-CoV-2 spike trimeric IgG with DiaSorin Trimeric spike IgG immunoassay on Liaison XL (DiaSorin, Saluggia, Italy) [3]. Results exceeding the upper linearity limit of both immunoassays (i.e., >25,000 kBAU/L for anti-RBD total antibodies and >108,160 kBAU/L for anti-trimeric spike IgG, respectively) were excluded, in order to allow for more accurate estimation of the relative post-booster increase in anti-SARS-CoV-2 antibodies. The measurement of anti-SARS-CoV-2 spike trimeric IgG antibodies was chosen since they sufficiently differ from anti-RBD antibodies, where a good correlation is obviously anticipated. Statistical analysis was carried out with Analyse-it (Analyse-it Software Ltd., Leeds, UK). All participants signed an informed consent for participating to the study, which was conducted in accordance with the Declaration of Helsinki and approved by the Ethics Committee of Verona and Rovigo Provinces (59COVIDCESC; 3 November 2021).

## 3. Results

The final study population consisted in 266 healthcare workers (median age, 46 years and interquartile range (IQR), 35–52 years; 168 females). The main results of this study are shown in Figure 1.

After receiving the BNT162b2 vaccine booster, the serum concentration increased by a median of 31 folds (IQR, 16–53), from 463 (IQR, 271–882) kBAU/L to 15,280 (IQR, 11,773–19,447) kBAU/L for anti-SARS-CoV-2 RBD total antibodies, and by a median of 22 (IQR, 11–43) folds, from 662 (IQR, 339–1255) kBAU/L to 13,624 (IQR, 9939–19,019) kBAU/L for anti-SARS-CoV-2 spike trimeric IgG, respectively. The absolute values of anti-SARS-CoV-2 antibodies assayed with the Roche immunoassay were predictably much higher than those obtained with the DiaSorin method, since the former technique measures total anti-SARS-CoV-2 antibodies, whilst the latter only detected IgG. A highly significant Spearman′s correlation was found between the relative increase (i.e., ratio of post-booster and pre-booster serum values) of anti-SARS-CoV-2 RBD total antibodies and anti-SARS-CoV-2 spike trimeric IgG (r = 0.86; 95%CI, 0.83–0.89; *p* < 0.001), as shown in Figure 1. This correlation was described by the following equation:[Anti-SARS-CoV-2 spike trimeric IgG] = [Anti-SARS-CoV-2 total antibodies] × 0.73 + 2.70

Such correlation remained highly significant both in males (r = 0.88; 95%CI, 0.82–0.92; *p* < 0.001) and females (r = 0.85; 95%CI, 0.80–0.89; *p* < 0.001), as well as in subjects aged <50 (r = 0.86; 95%CI, 0.81–0.89; *p* < 0.001) or ≥50 (r = 0.85; 95%CI, 0.78–0.90; *p* < 0.001) years.

## 4. Discussion

The results of our analysis were aimed to complement those published earlier by Skrzat-Klapaczyńska and colleagues [1] on elucidating the effect of BNT162b2 booster immunization on anti-SARS-CoV-2 antibodies. For this purpose, we measured both total and IgG anti-SARS-CoV-2 antibodies, showing that both classes of antibodies considerably increased (by 31 and 22-fold, respectively) 1 month after receiving the BNT162b2 vaccine booster. Besides providing an accurate measure of such increment in subjects with precisely measurable anti-SARS-CoV-2 antibodies values, we also demonstrated a very high correlation (i.e., *p* < 0.001) between the increase in anti-SARS-CoV-2 RBD total antibodies and anti-SARS-CoV-2 spike trimeric IgG, thus potentially reflecting a similar trend of humoral immune response despite the antigen target of these classes of immunoglobulins is different (i.e., trimeric spike protein vs. RBS). This is inherently important for demonstrating that even distinct class of antibodies, targeting different antigenic moieties on the spike protein, could be reliably used for purposes of monitoring COVID-19 vaccination.

## 5. Conclusions

The results of this study suggest that monitoring of post-BNT162b2 booster immunization for purposes of identifying “low responders” to vaccine boosters could be performed almost interchangeably with either total or IgG anti-SARS-CoV-2 antibodies.

## Figures and Tables

**Figure 1 vaccines-10-00890-f001:**
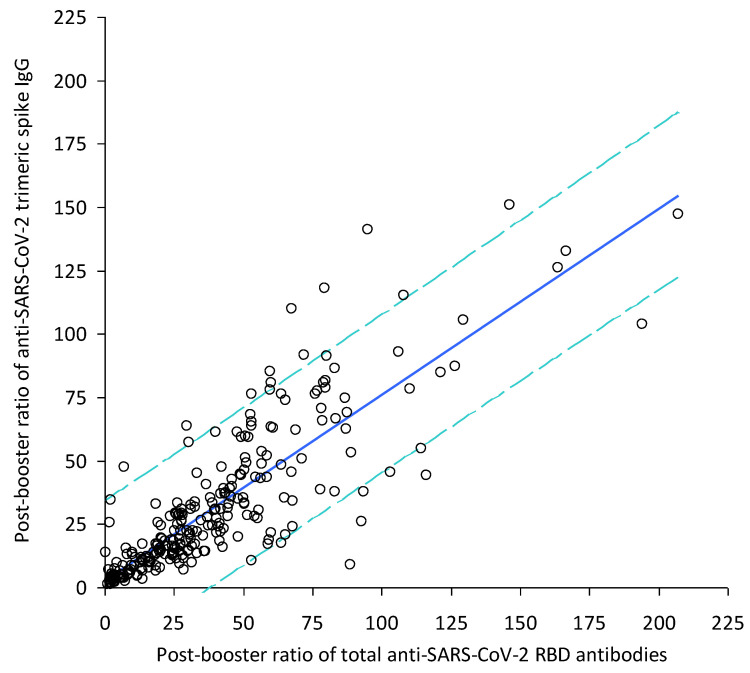
Correlation between relative increase (i.e., ratio of BNT162b2 post-booster and pre-booster serum values) of anti-SARS-CoV-2 total antibodies and anti-SARS-CoV-2 spike trimeric IgG.

## Data Availability

Data are available upon reasonable request to the corresponding author.

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
