# Peer review of "Correlation between Anti-SARS-CoV-2 Total Antibodies and Spike Trimeric IgG after BNT162b2 Booster Immunization"

_vaccines, 2022, doi:10.3390/vaccines10060890_

Round 1

Reviewer 1 Report

This work by Salvagno et al is a follow-up study to a previous paper published in Vaccines (Skrzat-Klapaczyńska, et al. 2022). The previous work found that a booster dose of the BNT162b2 mRNA vaccine would increase the percentage of people with high levels of S-RBD antibodies but not anti-n-protein IgG antibodies, which may indicate greater protection against both the disease and a severe course of COVID-19. In the current study, the authors further extended the scope by measuring the anti-SARS-CoV-2 RBD total antibodies and anti-SARS-CoV-2 spike trimeric IgG. The authors found that both of the two antibodies classes significantly increased after BNT162b2 vaccine booster immunization. Besides, they found the antibody level of the two classes are positively correlated, suggesting either total or IgG anti-SARS-CoV-2 antibodies can be used as an indicator for response to vaccine boosters. Overall, this study provided important value to the topic of measuring antibody level changes after a booster dose and the letter is well written.

I have only minor comments that did the authors also measure the IgG antibodies against the n-protein and whether that would further confirm the previous finding that no difference was found for the n-protein IgG antibodies? Besides, the authors might want to add a sentence to address why anti-SARS-CoV-2 spike trimeric IgG was chosen to measure.

Author Response

Reviewer: 1

This work by Salvagno et al is a follow-up study to a previous paper published in Vaccines (Skrzat-Klapaczyńska, et al. 2022). The previous work found that a booster dose of the BNT162b2 mRNA vaccine would increase the percentage of people with high levels of S-RBD antibodies but not anti-n-protein IgG antibodies, which may indicate greater protection against both the disease and a severe course of COVID-19. In the current study, the authors further extended the scope by measuring the anti-SARS-CoV-2 RBD total antibodies and anti-SARS-CoV-2 spike trimeric IgG. The authors found that both of the two antibodies classes significantly increased after BNT162b2 vaccine booster immunization. Besides, they found the antibody level of the two classes are positively correlated, suggesting either total or IgG anti-SARS-CoV-2 antibodies can be used as an indicator for response to vaccine boosters. Overall, this study provided important value to the topic of measuring antibody level changes after a booster dose and the letter is well written.

  • We are thankful to the referee for the globally favourable comments on our manuscript. We’ll do our best to improve it according to the referee’s suggestions.

I have only minor comments that did the authors also measure the IgG antibodies against the n-protein and whether that would further confirm the previous finding that no difference was found for the n-protein IgG antibodies? Besides, the authors might want to add a sentence to address why anti-SARS-CoV-2 spike trimeric IgG was chosen to measure.

  • ANSWER: We did not measured anti-N antibodies since this was not included in the original protocol submitted to the EC, nor it seems to have substantial meaning in this context since we excluded subjects with incident SARS-CoV-2 infection (this has been clarified, as follows: “Nucleic acid amplification tests were carried out at 2-4 weeks interval throughout the study period with Altona Diagnostics RealStar SARS-CoV-2 RT-PCR Kit (Altona Diagnostics GmbH, Hamburg, Germany) or Seegene Allplex SARS-CoV-2 Assay (Seegene Inc., South Korea), to rule out incident SARS-CoV-2 infections, that would lead to eliminating these subjects from the study, since booster vaccination is not normally given to people with history of recent infection”). We also did not measured anti-N antibodies because the Pfizer/Biontech vaccine does not elicit anti-N antibodies, whose measurement may hence be insignificant for the aim of our article. Therefore anti-N antibodies would be unmeasurable (waste of efforts and money) according to our study design.

As concerns the measurement of anti-SARS-CoV-2 spike trimeric IgG, this was chosen because it is a sufficiently different antibody from the anti-RBD. Inherently, there would not be too much sense in measuring two class of antibodies both targeting the same antigenic determinant (i.e., RBD), since they will expectedly give correlated data. We have included this explanation in the section “methods” (“The measurement of anti-SARS-CoV-2 spike trimeric IgG antibodies was chosen since they sufficiently differ from anti-RBD antibodies, where a good correlation is obviously anticipated”)

Reviewer 2 Report

The authors could try to provide a mechanistic insight into this high correlation in the discussion section to make the studies more intact. 

Author Response

The authors could try to provide a mechanistic insight into this high correlation in the discussion section to make the studies more intact. 

  • ANSWER: Good point, thanks. Text revised as follows: “Beside providing an accurate measure of such increment in subjects with precisely measurable anti-SARS-CoV-2 antibodies values, we also demonstrated a very high correlation (i.e., p<0.001) between the increase of anti-SARS-CoV-2 RBD total antibodies and anti-SARS-CoV-2 spike trimeric IgG, thus potentially reflecting a similar trend of humoral immune response despite the antigen target of these immunoglobulin classes is different (i.e., trimeric spike protein vs. RBS). This is inherently important for demonstrating that even distinct class of antibodies, targeting different antigenic moiety on the spike protein, could be reliably used for purposes of monitoring COVID-19 vaccination”.

Reviewer 3 Report

The manuscript of Salvagno and Lippi describes a correlation between anti-SARS-CoV-2 total RBD antibodies and spike trimeric IgG after 1 month of BNT162b2 booster immunization of 266 healthy healthcare workers. The results suggest that both methods are useful for identification of “low responders” to immunization.

Comments:

-   The sample population description can be enhanced. Example using a diagram, including age, gender.

-   The health care workers had SARS-COVID-2 infection history before or after the immunization? This information should be added to the diagram.

-   It is also clear that the sample population have more females than males. Exist some descriptions that the humoral response after BNT162b2 immunization can be age- and gender-dependent (example:  10.3390/microorganisms9081725). So, my suggestion is the analysis of the results of each gender group in separate.

-   It is also not clear in the manuscript why is important a correlation between the 2 different methods? Please add this information.

Author Response

Reviewer: 3

The manuscript of Salvagno and Lippi describes a correlation between anti-SARS-CoV-2 total RBD antibodies and spike trimeric IgG after 1 month of BNT162b2 booster immunization of 266 healthy healthcare workers. The results suggest that both methods are useful for identification of “low responders” to immunization.

-   The sample population description can be enhanced including age, gender.

  • ANSWER: Good point, thanks. Done, as follows: “The final study population consisted in 266 healthcare workers (median age, 46 years and interquartile range (IQR), 35-52 years; 168 females)”. See also below for additional data stratified by age and sex.

-   The health care workers had SARS-COVID-2 infection history before or after the immunization? This information should be added to the diagram.

  • ANSWER: Good point, thanks. Description added, as follows: “Nucleic acid amplification tests were carried out at 2-4 weeks interval throughout the study period with Altona Diagnostics RealStar SARS-CoV-2 RT-PCR Kit (Altona Diagnostics GmbH, Hamburg, Germany) or Seegene Allplex SARS-CoV-2 Assay (Seegene Inc., South Korea), to rule out incident SARS-CoV-2 infections, that would lead to eliminating these subjects from the study, since booster vaccination is not normally given to people with history of recent infection”.

-   It is also clear that the sample population have more females than males. Exist some descriptions that the humoral response after BNT162b2 immunization can be age- and gender-dependent (example:  10.3390/microorganisms9081725). So, my suggestion is the analysis of the results of each gender group in separate.

  • ANSWER: Done, as follows: “Such correlation remained highly significant both in males (r=0.88; 95%CI, 0.82-0.92; p<0.001) and females (r=0.85; 95%CI, 0.80-0.89; p<0.001), as well as in subjects aged <50 (r= 86; 95%CI, 0.81-0.89; p<0.001) or ≥50 (r=0.85; 95%CI, 0.78-0.90; p<0.001) years”.

-   It is also not clear in the manuscript why is important a correlation between the 2 different methods? Please add this information.

  • ANSWER: Good point: Text modified as follows: “Beside providing an accurate measure of such increment in subjects with precisely measurable anti-SARS-CoV-2 antibodies values, we also demonstrated a very high correlation (i.e., p<0.001) between the increase of anti-SARS-CoV-2 RBD total antibodies and anti-SARS-CoV-2 spike trimeric IgG, thus potentially reflecting a similar trend of humoral immune response despite the antigen target of these immunoglobulin classes is different (i.e., trimeric spike protein vs. RBS). This is inherently important for demonstrating that even distinct class of antibodies, targeting different antigenic moiety on the spike protein, could be reliably used for purposes of monitoring COVID-19 vaccination”.

Reviewer 4 Report

This study used 2 commercial quantitative serological  assays, detecting antibodies to the surface (S) antigen of the SARS-CoV-2 virus, to measure the increase in antibody titers after a booster dose of SARS-CoV-2 vaccine compared to the titer before this dose. These 2 assays are different in their format with one detecting total antibodies (IgG, IgA and IgM) against the RBD part of the S antigen (Roche test) and the other only IgG but against the trimeric conformation of total S antigen (DiaSorin test). They therefore do not detect the same antibodies. However, the ratios of titers after and before vaccine dose were well correlated for each subject between the 2 assays. Despite this, the median rate of rise in titer with the Roche assay is greater than that obtained with the DiaSorin assay. The authors could examine the significance of this difference, and discuss it.

The discussion is also unclear about what their results bring.

For example they compared their data expressed as a quantitative ratio without  any notion of what they consider a high or low ratio with another paper using a titer threshold to stratify high from low responders. The authors should specify how their study complements the previous one.

The same comment applies to the conclusion. They stated that :

"The results of this study suggest that monitoring of post-BNT162b2 booster immunization for purposes of identifying “low responders” to vaccine boosters could be performed..."

But they did not explain how they could identify "low responders" with their methods.

Author Response

Reviewer: 4

This study used 2 commercial quantitative serological  assays, detecting antibodies to the surface (S) antigen of the SARS-CoV-2 virus, to measure the increase in antibody titers after a booster dose of SARS-CoV-2 vaccine compared to the titer before this dose. These 2 assays are different in their format with one detecting total antibodies (IgG, IgA and IgM) against the RBD part of the S antigen (Roche test) and the other only IgG but against the trimeric conformation of total S antigen (DiaSorin test). They therefore do not detect the same antibodies. However, the ratios of titers after and before vaccine dose were well correlated for each subject between the 2 assays.

  • We are thankful to the referee for the globally favourable comments on our manuscript. We’ll do our best to improve it according to the referee’s suggestions.

Despite this, the median rate of rise in titer with the Roche assay is greater than that obtained with the DiaSorin assay. The authors could examine the significance of this difference, and discuss it.

  • ANSWER: Very good point, thanks. Manuscript modified accordingly, as follows: “The absolute values of anti-SARS-CoV-2 antibodies assayed with the Roche immunoassay was predictably much higher than those obtained with the DiaSorin method, since the former technique measures total anti-SARS-CoV-2 antibodies, whilst the latter only detected IgG”.

For example they compared their data expressed as a quantitative ratio without  any notion of what they consider a high or low ratio with another paper using a titer threshold to stratify high from low responders. The authors should specify how their study complements the previous one.

  • ANSWER: Good point, thanks. Explanation of low responders is given in the answer to the point below.

"The results of this study suggest that monitoring of post-BNT162b2 booster immunization for purposes of identifying “low responders” to vaccine boosters could be performed..." But they did not explain how they could identify "low responders" with their methods.

  • ANSWER: Good point, thanks. We have included this comment in the article “The results of this study suggest that monitoring of post-BNT162b2 booster immunization for purposes of identifying “low responders” to vaccine boosters (i.e., those with antibodies level below a predicted threshold of protection such as that identified by Feng et al., i.e., 264 and 506 kBAU/L for anti-spike and anti-RBD antibodies, respectively) (new ref. 4: Feng S, Phillips DJ, White T, Sayal H, Aley PK, Bibi S, Dold C, Fuskova M, Gilbert SC, Hirsch I, Humphries HE, Jepson B, Kelly EJ, Plested E, Shoemaker K, Thomas KM, Vekemans J, Villafana TL, Lambe T, Pollard AJ, Voysey M; Oxford COVID Vaccine Trial Group. Correlates of protection against symptomatic and asymptomatic SARS-CoV-2 infection. Nat Med 2021;27(11):2032-2040)…”.

Round 2

Reviewer 3 Report

The authors reviewed according to the comments and suggestions done, therefore the manuscript in the present form is acceptable for publication.